

# How Remote-Sensing Evapotranspiration Data Improve Hydrological Model Calibration in a Typical Basin of Qinghai-Tibetan Plateau Region

Jinqiang Wang[1], Ling Zhou[1], Chi Ma[1], Wenchao Sun[1]

[1]Beijing Key Laboratory of Urban Hydrological Cycle and Sponge City Technology, College of Water Sciences, Beijing Normal University, Beijing 100875, China

*Correspondence to*: Wenchao Sun (sunny@bnu.edu.cn)





**Abstract.** Many rivers in the East Asian Monsoon region originates from the Qinghai-Tibet Plateau (QTP), which provide
huge amount of fresh water resources for downstream counties. As a region characterized by high altitude and cold weather,
distributed hydrological modelling provide valuable knowledge about water cycle and cryosphere of the QTP. However, the
lack of streamflow data restricts the application of hydrological models in this data-sparse region. Previous studies have
demonstrated the possibility of using remote sensing evapotranspiration (RS-ET) data to improve modelling. However, in the
QTP, the mechanisms driving such improvements haven't been understood thoroughly. In this study, such driving mechanisms
were explored through the rainfall-runoff modelling of the Soil and Water Assessment Tool (SWAT) in the Yalong River
Basin of the QTP. Three experiments of model calibrations were conducted using streamflow data at the basin outlet, basins
averaged RS-ET data of the Global Land Evaporation Amsterdam Model (GLEAM), and the combination of the both data,
under the framework of the Generalized Likelihood Uncertainty Analysis (GLUE). The results show that compared with
calibration using streamflow data solely, the Nash-Sutcliffe Efficiency of simulated streamflow at 50% quantiles for the
calibration using both of streamflow and RS-ET data increased from 0.71 to 0.81 in the calibration period, while in the
validation period improved from 0.75 to 0.84, and more observations are embraced by the uncertainty bands. Similar
improvements are also found for the ET estimates. Comparison of parameter posterior distributions among the three
experiments demonstrated that calibration using both types of observations could increase the number of parameters that
posterior distributions are different from assumed uniform prior distribution, indicating the degree of equifinality was reduced.
A more comprehensive parameter sensitivity analysis by the Sobol' method were also conducted for reasoning the differences
among the three calibrations. Although the number of the detected sensitive parameters are almost same, the sensitive
parameter detected based on both types of observations covers surface runoff generation, snow-melting, soil water movement
and evaporation processes, while using single type of observations, the identified sensitive parameters are only the ones related
the hydrological processed quantified by the observations. From the aspects of model performance and parameter sensitivity,
it is demonstrated that not only the model output performs better, but also the characteristics of water cycle are captured more
effectively, highlighting the necessity of incorporating RS-ET data for hydrological model calibration in the QTP. Moreover,
adopting observations or information about soil property or snow-melting processes to make more reasonable estimates of
parameter distribution could further reduce simulation uncertainty under the calibration strategies proposed in this study.

# 1 Introduction

The Qinghai-Tibetan Plateau (QTP), known as "Asia's water tower," is a critical source of freshwater for many rivers, such as
the Yangtze River, Yellow River and Mekong River, which provide water resources for hundreds of millions of people residing
in the river basins (Li et al.,2014; Wang et al., 2018). In recent years, global climate change and intensified human activities
have significant impacts on the water resources in the region (Yao et al., 2019), which highlight the importance of examining
the changes in spatiotemporal dynamics of water cycles at basin scale (Huang et al., 2020; Li et al., 2014). Hydrological models
are useful tools for simulating rainfall-runoff processes and then provide valuable knowledges for sustainable water resources



managements (Huang et al., 2022). However, due to the lack of streamflow data for parameter calibration, the application of hydrological models in QTP are limited.

Traditionally, hydrological models are calibrated and validated against the measured streamflow data at the basin outlet (Becker et al., 2019; Dembélé et al., 2020a; Zhang et al., 2023). However, many studies show that the information contained in the streamflow data cannot completely capture the characteristics of internal hydrological processes of a specific basin (McDonnell et al., 2007). The evapotranspiration (ET) is such a process, which describes the phenomenon of water returning to the atmosphere from land surface, and can affect the processes of soil moisture, and runoff generation (Rajib et al., 2018; Shah et al., 2021; Zhang et al., 2020). Previous studies demonstrated that calibrating hydrological models only by measured streamflow may not gain reasonable estimates of model parameters and be a major cause of parameter equifinality (Dembélé et al., 2020b; López et al., 2017) and incorporating ET observations into model calibration may help to improve model simulation (Herman et al., 2018). RS-ET data products have been adopted by many researchers for calibrating hydrological models (Liu et al. 2022; Meyer Oliveira et al., 2021; Willem Vervoort et al., 2014; Zhang et al., 2021), due to its wide spatial coverage, and relatively higher spatial resolution, compare with ground observation. Immerzeel and Droogers (2008) were among the early pioneers for such studies and suggested the calibration of hydrological models using MODIS RS-ET data could lead to a better constrain of model. In the QTP, China, Huang et al. (2020) found that it is possible to use PML-V2 RS-ET data to calibrate Xinanjiang model (Zhao 1980), allowing for the estimation of daily and monthly runoff time series in ungauged or sparsely basins. It is evident that RS-ET data has been recognized as being effective in the calibration of hydrological models in ungauged basins. Gupta et al. (2006) argued that a hydrological model can be considered as being well-calibrated if fulfilling three necessary conditions: Firstly, the input-state-output behavior of the model is consistent with the measurements of catchment behavior; Secondly, the model predictions are accurate and precise; Thirdly, the model structure and behavior are consistent with a current hydrologic understanding of reality. Previous studies about using RS-ET for hydrological model calibration have explore the first and second conditions well by model calibration and validation. However, the value of RS-ET data in improving the understanding of model structure and behavior has not been understood thoroughly, especially in the data-sparse QTP, for which the characteristics of water cycle are quite unique, due to high altitude and cold climate.

Parameter Sensitivity analysis (SA), a method by continuously disturbing parameters in a model to evaluate the effects of the parameters' change on model outputs and state variables, can be used to reveal the key parameters that influence the hydrological cycle (Razavi and Gupta, 2015; Razavi et al., 2021; Song et al., 2015), and gain knowledge about the hydrological cycle under specific model structure (Mai et al., 2022). The Sobol' method, a global sensitivity analysis method based on variance decomposition, can provide quantitative estimates about the sensitivity of single parameter and parameter interaction for highly nonlinear models (Khorashadi Zadeh et al., 2017; Sobol', 1993). Basijokaite et al. (2021) used Sobol' SA to examine parameter sensitivity of HYMOD hydrological model in 30 watersheds in California, USA. It is revealed that the identified difference of parameter sensitivity could reflect the difference in basin characteristics. Zhang et al. (2013) applied Sobol' SA for distributed hydrological model of SWAT in the Yichun Basin, China, and successfully identified that, compared with the



dry year, the parameter interactions contribute much more to parameter sensitivity under wet year. Previous studies demonstrated that there are varying factors that affect hydrological simulations within a watershed, and understanding these variations using Sobol' method can help to quantify the contribution of each parameter and parameter interaction to model output and subsequently lead to a deeper insight about driving mechanism of water cycle at the basin scale.

Based on these knowledges, the objective of this study is to fully explore the value of incorporating RS-ET data into

hydrological model calibration in the QTP region through SWAT modeling in the Yalong River Basin. The SWAT was calibrated by GLUE based on streamflow data, RS-ET data, and both of streamflow and RS-ET data, respectively, and then the difference in simulation accuracy, uncertainty and posterior parameter distribution among the above-mentioned three experiments of model calibration were examined. Thereafter, mechanisms leading to the differences were inspected using the Sobol' method by quantitively analyzing the sensitivity of single parameter and parameter interaction. The findings of this

study are expected to achieve a more comprehensive understanding about feasibility of using RS-ET data for hydrological model calibration and provide guidance for further reducing simulation uncertainty in the data-sparse QTP.

## 2 Materials and Methods

### 2.1 Study area

The Yalong River, which originates from the Bayan Har Mountain in QTP, is the largest tributary of the Jinsha River, which

is the headwater river of the Yangtze River. In this study, the hydrological simulation was conducted in the upstream region of the Ganzi Hydrological Station (Fig. 1). The study area is located in the southeastern of the QTP, spans from the northwest to southeast and the altitude of the basin is 3,400 m - 6,021 m, with the total length of the mainstream is around 690 km and the whole basin area is 32,535 km$^2$. Affected by the plateau monsoon climate, the study area has distinct plateau characteristics such as long winters and short summers, significant diurnal temperature variation, high solar radiation, and a pronounced dry

and wet season, additionally, which are typical of plateau environments. The basin receives an average annual precipitation of approximately 530 mm, mostly occurring between June and September. Due to the high altitude, the study area experiences an extended snowfall period of over 9 months. The average annual temperature varies from -4.9℃ to 7.8℃. Runoff primarily originates from precipitation, with additional contributions from groundwater and snowmelt (Huang et al., 2020; Kang et al., 2001). The dominant soil types are Plateau Meadow, and the major land use and land cover types (LULC) are shrub meadow.

**Figure 1.** (a) Topography, (b) Subbasin, (c) Landcover and (d) Soil type of the upstream region of the Ganzi hydrological station in the Yalong River.

## 2.2 Rainfall-runoff modelling and data source

### 2.2.1 SWAT model

SWAT is a widely used physically based hydrological model developed by the United States Department of Agriculture – Agricultural Research Service (USDA-ARS). It simulates hydrological process, transportation and transformation of pollutants





at the basin scale. Based on the river network derived from digital elevation model, the study area being modeled are partitioned into many subbasins to account for the spatial heterogeneity of hydrological processes within a basin. The runoff generation processes are modeled for each subbasin separately and the river flow movement are modelled based on river network from upstream to the downstream directions. Full details of SWAT model are referred to Arnold et al. (1998). The SWAT model simulates ET by considering the combined effects of river, soil, and vegetation surface evaporation, as well as plant transpiration. The SWAT model employs the Penman-Monteith method, Priestley-Taylor or Hargreaves method, to indirectly estimate actual ET. In this study, the Penman-Monteith method was utilized for calculating ET.

### 2.2.2 Data sources and model setting

There are several kinds of datasets required to build the SWAT model (Table 1), including GIS data of DEM, landuse and landcover, soil type within the basin, and meteorological input data of precipitation and air temperature. Spatial and temporal variations of precipitation and air temperature are high in this region, which cannot be fully captured by sparse ground gauging system. Therefore, two grid data products based on satellite observations are adopted as input forcing data. The Multi-Source Weighted Ensemble Precipitation (MSWEP) V2.8, which is a global precipitation dataset generated by merging gauge, satellite and model simulation data, with long time scale (1979 - near present) and high temporal (3h, day, month) and spatial (0.1°) resolution (Beck et al., 2019). The gridded temperature dataset, obtained from Multi-Source Weather (MSWX), was downscaled using high-resolution climatology data based on ERA5. The daily maximum and minimum temperature values from MSWX were used in this study. For the model calibration based on ET data, the GLEAM 3.5a grid dataset was utilized. It provides estimates of land surface evaporation, were developed and provided by researchers Miralles et al. (2011). The datasets have a spatial resolution of 0.25° and cover a long-time scale from 1980 to near present, with daily, monthly, and yearly time scales available. In this study, the GLEAM 3.5a monthly dataset of actual ET was used for model calibration. Meanwhile, measured streamflow data of Ganzi station were also used to calibrate and validate the model.

For SWAT modelling, the calibration and validation period were set as years 2001-2005 and 2006-2010, respectively. The studied basin was discretized into 29 subbasins (Fig. 1b) based on river network and the model was run at monthly scale. For each subbasin, the area averaged daily precipitation, daily maximum temperature, and monthly ET was computed form the grid datasets. Based on literature review (Neitsch, et al., 2011; Sun, et al., 2017), 28 parameters related streamflow and ET of SWAT modelling were selected for calibration and sensitivity analysis (Table 2).






**Table 1.** Overview of the Modelling Data Sets

| Type of Data | Data Product | Spatial Resolution | Temporal resolution | Source |
|---|---|---|---|---|
| DEM | Shuttle Radar Topography Mission | 90×90 m | - | http://srtm.usgs.gov |
| Land Use and Land Cover | Institute of Geographic Sciences and Natural Resources Research, Chinese Academy of Sciences | 1×1 km | - | http://www.igsnrr.cas.cn/ |
| Soil | Nanjing Institute of Soil Science, Chinese Academy of Sciences | 1×1 km | - | http://www.issas.cas.cn/ |
| Precipitation | MSWEP | 0.1° | Daily | http://www.gloh2o.org/mswep |
| Temperature | MSWX | 0.1° | Daily | http://www.gloh2o.org/mswx |
| RS-ET | GLEAM 3.5a | 0.25° | Monthly | https://www.gleam.eu |
| Streamflow | Hydrological Year Book | - | Monthly | Ministry of Water Resources, China |


**Table 2.** Parameters of SWAT being calibrated in this study

| Hydrological | No. | Parameter Name | Parameter Description | Unit | Min | Max |
|---|---|---|---|---|---|---|
| Runoff generation | 1 | CN2 | Runoff curve number multiplicative factor | - | 35 | 98 |
| | 2 | HRU_SLP | Average slope steepness | m m$^{-1}$ | 0 | 1 |
| | 3 | LAT_TTIME | Lateral flow travel time | days | 0 | 180 |
| | 4 | SLSUBBSN | Average slope length | m | 10 | 150 |
| | 5 | OV_N | Manning's 'n' value for the overland flow | - | 0.91 | 30 |
| | 6 | SURLAG | Surface runoff lag time | days | 0.05 | 24 |
| Groundwater | 7 | GW_DELAY | Groundwater delay time | days | 0 | 500 |
| | 8 | ALPHA_BF | Baseflow alpha factor | days | 0 | 1 |
| | 9 | GWQMN | Threshold depth of water in the shallow aquifer required for return flow to occur | mm | 0 | 5000 |
| | 10 | GW_REVAP | Groundwater 'revap' coefficient | - | 0.02 | 0.2 |
| | 11 | REVAPMN | Threshold depth of water in the shallow aquifer for "revap" to occur | mm | 0 | 500 |
| | 12 | RCHRG_DP | Deep aquifer percolation fraction | - | 0 | 1 |
| Soil Water Movement | 13 | SOL_BD | Moist bulk density | mg m$^{-3}$ | 0 | 1 |
| | 14 | SOL_AWC | Available water capacity of the soil layer | - | 0 | 2000 |
| | 15 | SOL_K | Saturated hydraulic conductivity | mm h$^{-1}$ | 0 | 0.25 |
| Main Channel Processes | 16 | CH_N2 | Manning's 'n' value for the main channel | | 0.01 | 0.3 |
| | 17 | CH_K2 | Effective hydraulic conductivity in main channel alluvium | mm h$^{-1}$ | 0.01 | 500 |
| | 18 | CH_COV1 | Channel cover factor | - | 0 | 1 |
| | 19 | ALPHA_BNK | Baseflow alpha factor for bank storage | days | 0.05 | 0.6 |
| ET | 20 | TLAPS | Temperature lapse rate | °C km$^{-1}$ | -10 | 10 |
| | 21 | CANMX | Maximum canopy storage | mm | 0 | 100 |
| | 22 | ESCO | Soil evaporation compensation factor | - | 0 | 1 |
| | 23 | EPCO | Plant uptake compensation factor | - | 0 | 1 |
| Snow Melting | 24 | SFTMP | Snowfall temperature | °C | -20 | 20 |
| | 25 | SMTMP | Snowmelt base temperature | °C | 0 | 20 |
| | 26 | SMFMX | Maximum melt rate for snow during year | mm °C$^{-1}$day$^{-1}$ | 0 | 20 |
| | 27 | SMFMN | Minimum melt rate for snow during year | mm °C$^{-1}$day$^{-1}$ | 0 | 20 |
| | 28 | TIMP | Snowpack temperature lag factor | - | 0 | 1 |





### 2.3 Design of experiments

In order to demonstrate the differences of model performance and parameter sensitivity constrained between calibration based
on ground observed streamflow data and RS-ET data. Three experiments were conducted: In Experiment I, the model was calibrated using observed streamflow data only. In Experiment II, the model was calibrated using RS-ET data only. In Experiment III, the model was calibrated and validated using both streamflow and RS-ET data. For above mentioned three calibration experiments, the model performance of streamflow and ET for the calibration and validation were all evaluated. Meanwhile, the uncertainty bands of streamflow or ET simulation of each experiment were also computed to quantify
simulation uncertainty. Subsequently, sensitivity analysis was carried out based on the calibration data used in the three experiments, respectively, for the purpose to examining the mechanism driving the differences of model behaviors among the SWAT modeling. The details of model calibration, uncertainty analysis and sensitivity methods are given in the coming section. The setting of these methods is kept same to ensuring that the differences in modelling accuracy, simulation uncertainty and identified parameter sensitivity only comes from the differences in the calibration data.

### 170 2.4 Model calibration, uncertainty evaluation and sensitivity analysis method

#### 2.4.1 Model calibration and uncertainty evaluation method

The essential assumption of GLUE is that there is no single set of best parameter values and the parameter sets for which the value of likelihood measure quantifying model performance is all higher than certain threshold should be treated as behavioral parameter sets and be included in the ensemble simulation (Beven and Binley, 1992). In this study GLUE was applied for
automatic calibration and simulation uncertainty analysis and was conducted as follows:

Firstly, for the 28 parameters being calibrated, based on the prior range in Table 2 and assumption of uniform distribution, 10000 sets of model parameter were randomly generated by Latin hypercube sampling and each parameter set was used to run model.

Secondly, to identify the behavioral parameter sets, a likelihood measure must be selected. For the Experiment I, the Nash-
Sutcliffe Efficiency (NSE) of simulated streamflow at Ganzi station $NSE_Q$ was used as the likelihood:

$$L_y(\theta|Y)_{EXP_I} = NSE_Q = 1 - \frac{\sum(Q_{\text{obs,i}}-Q_{\text{sim,i}})^2}{\sum(Q_{\text{obs,i}}-Q_{\text{obs,avg}})^2} \tag{1}$$

where $L_y(\theta/Y)_{EXP\,I}$ represents the likelihood of a specific parameter set $\theta$ in Experiment I, $Q_{obs,i}$ and $Q_{sim,i}$ stand for measured and simulated streamflow at the time step $i$, respectively, $Q_{obs,avg}$ represents the average of observed streamflow. For the Experiment II, the NSE of simulated basin-averaged ET $NSE_{ET}$ was computed as the likelihood:

$$L_y(\theta|Y)_{EXP_{II}} = NSE_{ET} = 1 - \frac{\sum(ET_{\text{obs,i}}-ET_{\text{sim,i}})^2}{\sum(ET_{\text{obs,i}}-ET_{\text{obs,avg}})^2} \tag{2}$$

where $L_y(\theta/Y)_{EXP\,II}$ is likelihood of a specific parameter set $\theta$ in Experiment II simulated, $ET_{obs,i}$ and $ET_{sim,i}$ stand for measured and simulated ET at the $i$th time step, which are basin averaged value, and $ET_{obs,avg}$ represents the temporal average of observed ET. The basin-averaged observed or simulated ET was calculated as follows:





$$ET_j = \frac{1}{A_T} \sum_{i=1}^{n} A_i ET_{ij} \tag{3}$$

where $ET_j$ is the basin-averaged observed or simulated ET at the time step $j$, $A_T$ is the total area of the basin, $A_i$ is the area of sub-basin $i$, $ET_{ij}$ is the area averaged observed or simulated ET for subbasin $i$ in the $j$th time step, and $n$ is the number of subbasins. For Experiment III, the performance of simulated streamflow and ET are integrated to build the likelihood measure. The NSE of simulated streamflow and ET were computed in the same way as in Experiment I and II, respectively. Then the $NES_Q$ and $NSE_{ET}$ are combined as the likelihood:

$$L_y(\theta|Y)_{EXP_{III}} = 0.5 * NSE_Q + 0.5 * NSE_{ET} \tag{4}$$

which means in the likelihood measure, the weight of performance of simulated streamflow and ET are same. For each calibration, the parameter set for which the likelihood value is larger than 0.5 was treated as the behavioral parameter set. Lastly, after scaling the likelihood of all behavioral parameter set to 1, the cumulative distribution of simulated streamflow or ET were computed as:

$$P_t(Y_t < y) = \sum_{i=1}^{m} L_P [\theta_i|Y_{t,i} < y] \tag{5}$$

where $P_t(Y_t<y)$ is the cumulative probability of simulated streamflow or ET at the time step $t$ less than an arbitrary value of $y$, $L_p$ is the scaled likelihood of set $\theta_i$, m denote the total number of parameter sets satisfying the condition $Y_{t,i} < z$.

To quantify the model performance, for each calibration experiment, the NSE of simulated streamflow or ET corresponding to at the 50 quantiles of all-time step ($NSE_{50\%}$) are computed. To quantify the simulation uncertainty, simulated streamflow or ET corresponding to 5% and 95% quantiles of each time step were treated as the lower limit and upper limit of the uncertainty band of streamflow or ET, respectively. Then two indexes were calculated. The $P\_factor$ describes the percentage of observation embraced by the uncertainty band:

$$P\_factor = \frac{n_q}{n} * 100\% \tag{6}$$

where $n_q$ represents the number of measured observations falling within the uncertainty interval, and $n$ is the number of simulation time steps. The $R\_factor$ quantifies the average width of uncertainty band:

$$R\_factor = \frac{\frac{1}{n}\sum_{i=1}^{n} V_{s,upper}^i - V_{s,lower}^i}{\sigma_{obs}} \tag{7}$$

Where $V_{s,upper}^i$, $V_{s,lower}^i$ is the 95% and 5% quantiles of simulated variable at the time step $i$, $n$ is the number of simulation time steps, $\sigma_{obs}$ is the standard deviation of the observed variable. The simulation uncertainty is evaluated by the two indexes simultaneously. The possible ranges of $P\_factor$ and $R\_factor$ are 0 to 1 and 0 to positive infinity, respectively. A high value of $P\_factor$ combined with a low $R\_factor$ value indicates the simulation uncertainty is low

### 2.4.2 Sensitivity analysis

Distributed hydrological model can capture the complex hydrological processes occurring in a basin by making mathematical or physical assumptions that involve multiple parameters that generalize basin characteristics. However, not all parameters are equally important in hydrological simulations. Identifying the most influential parameters that affect hydrological simulations


within a specific basin is a crucial for understanding the water cycle of the basin. This can be accomplished through sensitivity

analysis of the parameters, which is expected to reveal the mechanism driving the differences in calibration using different

data in this study.

The Sobol' method is a global sensitivity analysis technique that utilizes variance decomposition to identify the key

parameters and their interactions for highly nonlinear models (Sobol', 1993; Khorashadi Zadeh et al., 2017). The total variance

of the model output can be separated into the variance caused by a single parameter and the variance caused by interaction

among multiple parameters. The proportion of variance caused by a single parameter to the total variance is called the first-

order sensitivity index (S1), which indicates the influence of parameter itself on the model output. The proportion of variance

caused by interaction between two or more parameters to the total variance is called the high-order sensitivity index, which

indicates the influence of interaction between two or more parameters on the model output. The sum of first-order and high-

order sensitivity index is called the total-order sensitivity index (ST), which indicates the total influence of parameters on the

model output. In most cases, S1 and second-order index (S2) make a large portion of parameter sensitivity. Therefore, in this

study, we calculated the S1, S2 and ST for sensitivity analysis. In the Sobol' method, suppose that a non-linear and non-

monotonic model can be represented by function $Y$:

$$Y = f(X) = f(X_1, \dots X_P) \tag{8}$$

where $Y$ represents the output of the model, $X$ represents the parameters of the model, and the variance $D(y)$ of function $f(X)$

can be decomposed into the variance caused by a single parameter and the variance caused by the interaction between multiple

parameters:

$$D(y) = \sum_i D_i + \sum_{i<j} D_{ij} + \sum_{i<j<k} D_{ijk} + \dots + D_{12\dots p} \tag{9}$$

where $D_i$ represents the impact of the $i_{th}$ parameter $X_i$ on the simulation results, $D_{ij}$ represents the impact of the interaction

between the $i_{th}$ parameter $X_i$ and the $j_{th}$ parameter $X_j$ on the simulation results, and then calculate the impact of the interaction

between a single parameter or multiple parameters on the model output according to their percentage contribution to the total

variance $D(y)$.

First-order index:

$$S_i = \frac{D_i}{D} \tag{10}$$

Second-order index:

$$S_{ij} = \frac{D_{ij}}{D} \tag{11}$$

Total-order index:

$$S_{Ti} = 1 - \frac{D_{\sim i}}{D} \tag{12}$$

where $S_i$ denotes the main effect of parameter $X_i$ to the output, $S_{ij}$ denotes the interaction between parameter $X_i$ and $X_j$, $S_{Ti}$

denotes the main effect and interaction of parameter $X_i$ with other parameters. The estimation of variance $D$, $D_i$, and $D_{\sim i}$ can

be obtained through the application of approximate Monte Carlo numerical integration (Sobol', 1993).





Sampling size is an important step in calculating sensitivity. Sobol' sequence, a popular quasi-random low-discrepancy sequence, can generate uniform samples of parameter space, Saltelli's scheme extends the Sobol' sequence to reduce the error rate of the calculation of sensitivity, the model simulates numbers M = N × (2 × D + 2), where N is the sample numbers, D is

the number of parameters, and M is the number of modelling. Since Sobol' sensitivity analysis requires a large number of sampling, we followed prior studies (Fu et al.2012; Zhang et al.2013) and sampled 4096 points in our research, i.e. M = 4096 × (2 × 28 + 2) = 237568 model evaluations. For each of the three experiment, the sensitivity analysis was conducted using the defined likelihood as the objective function to quantify the disturbance on model output.

### 3 Results

#### 3.1 Comparison of simulation accuracy and uncertainty among three experiments

Figure 2-4 show the simulated streamflow hydrograph corresponding to the Experiment I to III and the metrics of model performance were listed in Table 3. In Experiment I and III, for which streamflow data are used as calibration data, the temporal variation and magnitude of streamflow are reproduced well by the calibrated models and Experiment III outperforms Experiment I, judging from the $NSE_{50\%}$. In comparison, the Experiment II fails to capture the magnitude of temporal changes

in streamflow and the $NSE_{50\%}$ is much lower than the other two experiments, indicating only using RS-ET data cannot obtain reasonable estimates of parameter related to runoff generation. From the aspect of simulation uncertainty, uncertainties bands of Experiment I and III could contain a large portion of streamflow observations and Experiment III contains more, according to the values of $P\_factor$ in both of the calibration and validation period. It is also noticed that, in the view of $R\_factor$, the uncertainty band corresponding to Experiment I is wider than Experiment III, implying that the simulated streamflow by

behavioral parameter sets identified in Experiment III are more diverse.

For Experiment I to III, the simulated time series of ET are demonstrated in Fig. 5-7, respectively and the values of model accuracy and uncertainty are described in Table 4. For Experiment I, the $NSE_{50\%}$ for both of calibration and validation period are lower than 0 for ET simulation, indicating the calibration based on streamflow data solely cannot yield ET estimates with satisfactory performance in the studied basin, highlighting the necessity of incorporate ET information into hydrological model

calibration. Both of Experiment II and III made reasonable reproduction of the basin-averaged ET observed from remote sensing data. The $NSE_{50\%}$ corresponding to Experiment II, which only use RS-ET data for model calibration is slightly higher than Experiment III, for which model were calibrated based on both of RS-ET and streamflow data. Regarding the $P\_factor$ and $R\_factor$, the difference among the Experiment II and III are not significant, which means that the simulation uncertainties of the two experiments are in the similar level.



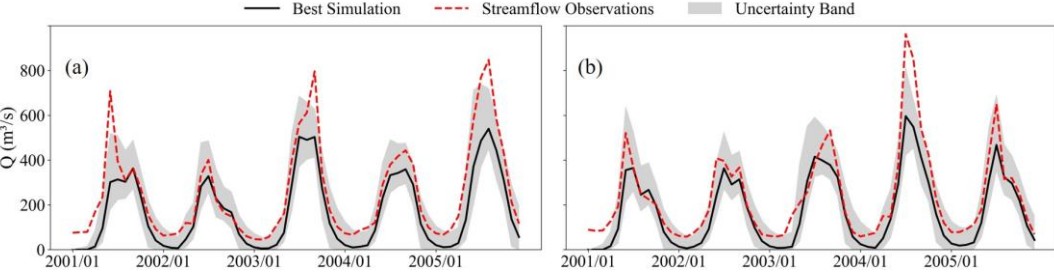

**Figure 2.** Simulated streamflow for the (a) calibration and (b) validation period corresponding to Experiment I.

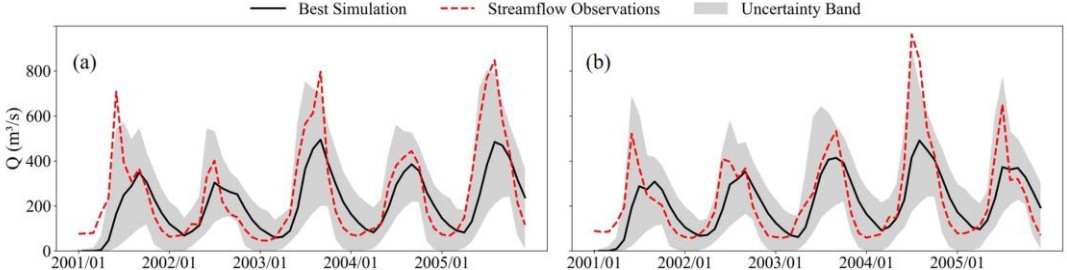

**Figure 3.** Simulated streamflow for the (a) calibration and (b) validation period corresponding to Experiment II.

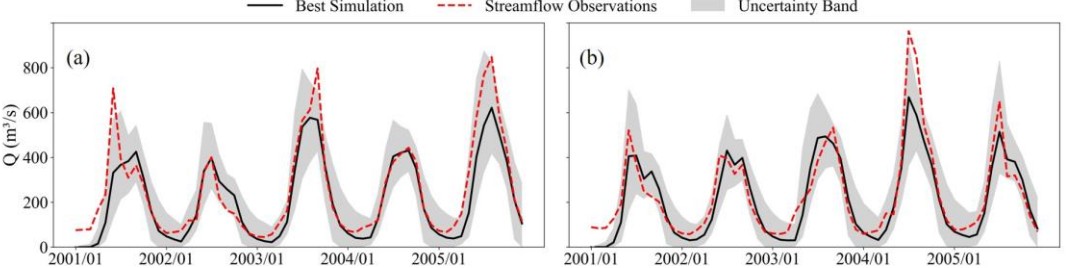

**Figure 4.** Simulated streamflow for the (a) calibration and (b) validation period corresponding to Experiment III.

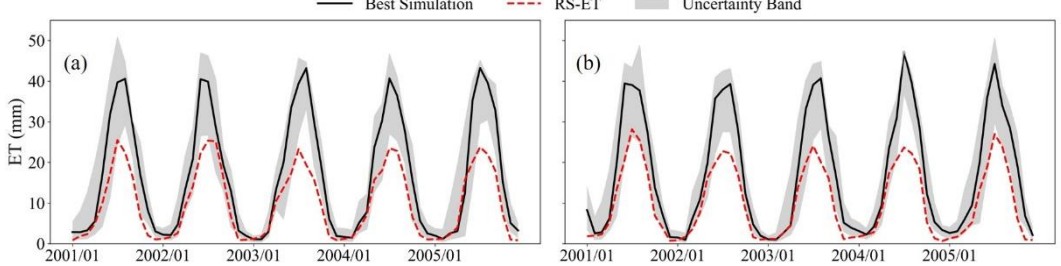

**Figure 5.** Simulated ET for the (a) calibration and (b) validation period corresponding to Experiment I.



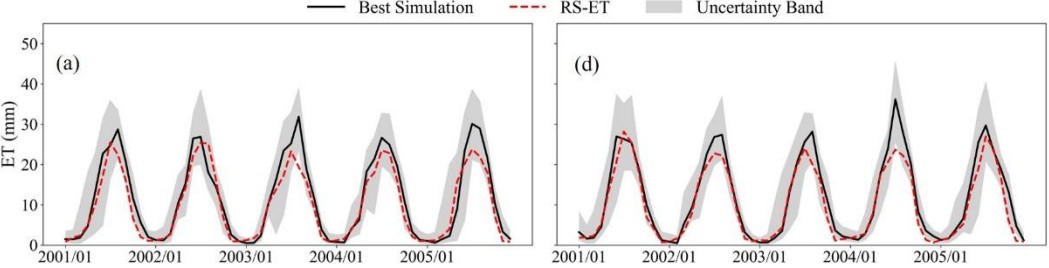

**Figure 6.** Simulated ET for the (a) calibration and (b) validation period corresponding to Experiment II.

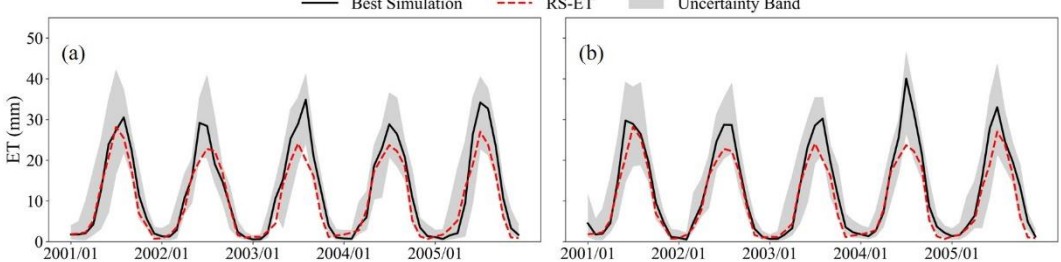

**Figure 7.** Simulated ET for the (a) calibration and (b) validation period corresponding to Experiment III.

**Table 3.** Accuracy and uncertainty for streamflow simulation

| | Experiment I | | Experiment II | | Experiment III | |
|---|---|---|---|---|---|---|
| | Calibration Period | Validation | Calibration Period | Validation | Calibration | Validation |
| $NSE_{50\%}$ | 0.70 | 0.75 | 0.52 | 0.54 | 0.81 | 0.84 |
| P_factor | 0.75 | 0.75 | 0.88 | 0.88 | 0.85 | 0.87 |
| R_factor | 0.86 | 0.92 | 1.42 | 1.51 | 1.1 | 1.17 |

**Table 4.** Accuracy and uncertainty for ET simulation

| | Experiment I | | Experiment II | | Experiment III | |
|---|---|---|---|---|---|---|
| | Calibration | Validation | Calibration | Validation | Calibration | Validation |
| $NSE_{50\%}$ | -0.22 | -0.09 | 0.84 | 0.88 | 0.79 | 0.79 |
| P_factor | 0.58 | 0.52 | 0.93 | 0.88 | 0.88 | 0.88 |
| R_factor | 1.42 | 1.35 | 1.2 | 1.14 | 1.18 | 1.18 |

### 3.2 Comparison of simulation accuracy and uncertainty among three experiments

Figure 8-10 demonstrated the posterior distribution of behavioral parameter sets for the three experiments, respectively. The deviation of posterior distribution from the initially assumed uniform distribution can be considered as indications of how the observed data used for calibration constrains parameters response surface. For the three experiments, the number of identified behavioral parameter set is 1663, 291 and 277, respectively, implying different observations restrict model behaviors in a different manner.





When calibrating using streamflow data only (Experiment I, Fig. 8), posterior distribution of parameters related to lateral flow (ALPHA_BNK and LAT_TTIME), surface runoff generation (CN2), and snow melting (SMTMP) are found to be different from priori uniform distribution. In Experiment II, for which the model was calibrated solely based on RS-ET data
(Fig. 9), parameters take on visible dissimilarity between assumed prior and identified posterior distribution are related to soil water movement and evaporation (SOL_AWC, SOL_K and ESCO), and snow melting (SFTMP and SMTMP). For the Experiment III using both streamflow and RS-ET data for calibration at the same time (Fig. 10), the detected parameters with different posterior distribution from uniform distribution includes the above-mentioned parameters and two additional parameters regarding to runoff generation (HRU_SLP and SLSUBBSN).

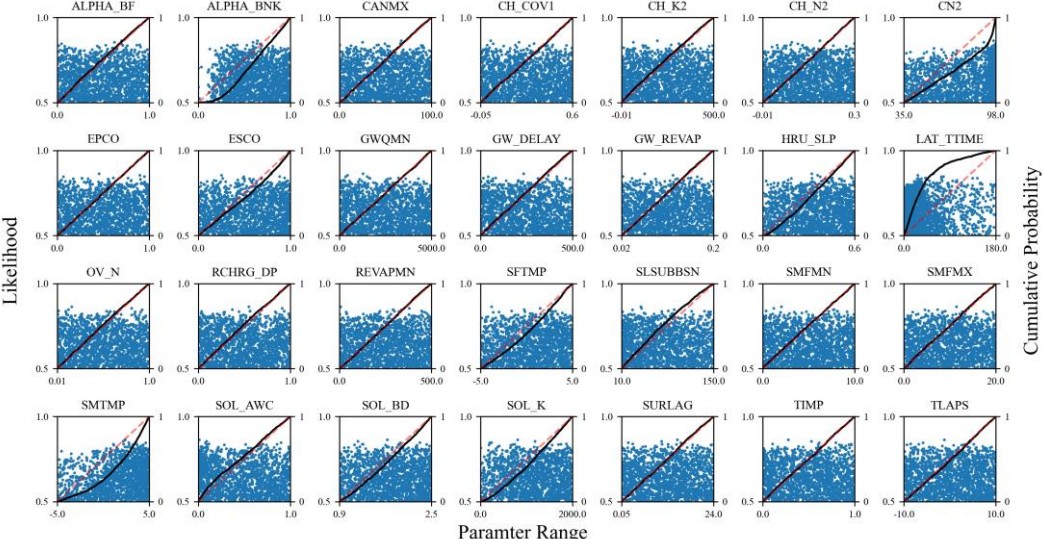

**Figure 8.** Posterior parameter distributions of Experiment I.





**Figure 9.** Posterior parameter distributions of Experiment II

**Figure 10.** Posterior parameter distributions of Experiment III.

## 3.3 Parameter sensitivity detected from the Sobol' method

Figure 11 demonstrates the results of Sobol's Sensitivity Analysis for the three experiments. The first-order index (S1) reflects the influences of an individual parameter on model output, and the total-order index (ST) reveals the impact of a single parameter plus it's all interaction with other parameters. The second-order index (S2) were also computed to show the interaction between two parameters on model output. Meanwhile the value of ST is larger than 0.1, or the value of S2 is larger than 0.01, the parameter or the interaction between two parameters are considered to be sensitive.



For Experiment I, which considers runoff as the output variable, there are five parameters that their variance contribution to the total model variance, i.e., ST, are larger than 0.1 and considered as being sensitive. More specifically, the parameter of ALPHA_BNK, LAT_TTIME, CN2, SMTMP and HRU_SLP, accounting for 34%, 26%, 13%, 12%, 10% of the total variance, respectively. They are related to the surface runoff generation, lateral flow, snow-melting processes. Furthermore, LAT_TTIME and ALPHA_BNK are found to have an interactive influence on the output, accounting for 1.6% of the total variance, which describes the movement of water in unsaturated zone and bank storage. When using RS-ET solely as calibration data (Experiment II), five parameters are also identified as sensitive: SMTMP, SOL_AWC, ESCO, SOL_K, HRU_SLP, accounting for 26%, 20%, 17%, 16% and 15% of the total variance, respectively. Two of the five parameters (SMTMP and HRU_SLP) are same as Experiment I and the other three (SOL_AWC, ESCO, SOL_K) are different, which describing the processes of soil water movement and evaporation. It is also found that two parameters related to snow-melting, i.e., SFTMP and SMTMP, have a non-negligible interactive influence on the output which account for 4.3% of the total variance. For Experiment III integrating both of streamflow and RS-ET data for calibration, six parameters are identified as sensitive: SMTMP, HRU_SLP, SOL_AWC, SOL_K, ESCO, CN2 and accounting for 24%, 15%, 14%, 14%, 12%, and 11% of the total variance, respectively. These six parameters were previously detected being sensitive in Experiment I or II. Additionally, it is found that two pairs of parameters, SFTMP and SMTMP, ESCO and TIMP have an interactive influence on the output which account for 3.3% and 1% of the total variance, respectively. These parameters describe snow-melting and soil water processes.







**Figure 11.** (a) S1 and ST and (b) S2 values for Experiment I; (c) S1 and ST and (d) S2 values for Experiment II; and (e) S1 and ST and (f) S2 values for Experiment III (parameter number is same as in Table 2).



## 4 Discussion

### 4.1 The value of adding RS-ET data for model calibration in QTP

In the three experiments of model calibration, all the settings of GLUE are the same, except the observation data, to ensure the differences of model simulation solely comes from the choice of calibration data. Our results show that when combing streamflow and RS-ET data for model calibration, the accuracy of simulated streamflow and ET are all higher and more observations are embraced by uncertainty band compared to calibration using streamflow data only. It is consistent with previous research by Herman et al. (2018) and Nijzink et al. (2018). Under the GLUE framework, the likelihood is assumed

to be a quantitative measure of the goodness of a parameter set to capture the characteristics of the hydrological processes. However, the hydrological variables for which the observations are used to compute the likelihood may be different from the hydrological variables being concerned by the modeler. A strong positive correlation of the likelihood and the accuracy of the simulated target hydrological variables are expected for a successful model simulation. To explore the mechanisms driving the difference of model simulation among the three experiments, a detailed analysis about the correlation between the likelihood

and the $NSE_Q$ or $NSE_{ET}$ was conducted for the behavioral parameter sets detected in each experiment (Fig. 12). When using streamflow only to calibrate the SWAT model (Figure. 12a), for the parameter sets with same likelihood, their $NSE_{ET}$ varies significantly, and the performance of $NSE_{ET}$ for many parameter sets are highly unsatisfactory. Similarly, when using the RS-ET data only (Figure. 12b), the accuracies of simulated streamflow among the behavior parameter sets may change considerably, although their likelihood are among the similar level. The streamflow is the ultimate output of the rainfall-runoff

processes within a basin, therefore streamflow data contains information about hydrological processes contribute to it. ET is one such process, but not the only one.  The results of Experiment I implies that the information contained in the streamflow data cannot fully capture the variation in ET. Meanwhile, as an internal variable of the runoff generation system, the ET data cannot completely reflect the magnitude and temporal pattern of changes in streamflow, it is the most possible reason that could explains the phenomenon in Fig. 12 b. When combining the streamflow and RS-ET data to calibrate the model (Fig. 12c

and 12d), compared with the Experiment I and II, the variations of $NSE_Q$ or $NSE_{ET}$ are much lower for the parameter set with same likelihood and lower limit of $NSE_Q$ or $NSE_{ET}$ increases as the increase of likelihood. All these facts indicate that the positive correlation between the likelihood and $NSE_Q$ or $NSE_E$ becomes stronger than the previous two experiments.





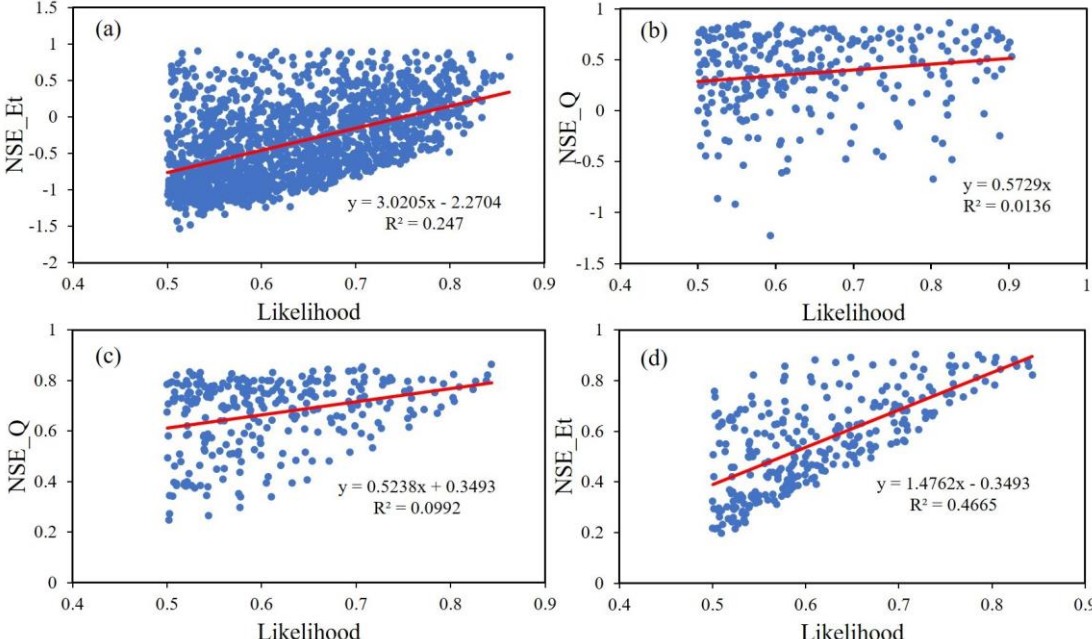

**Figure 12.** Likelihood of behavioural parameter sets in (a) Experiment I versus $NSE_{ET}$, (b) Experiment II versus $NSE_Q$, (c) Experiment III versus $NSE_Q$, and Experiment III versus $NSE_{ET}$.

The above discussion confirmed that incorporating RS-ET data with streamflow observations are effective to guarantee satisfactory model performance of ET and streamflow at the same time. Besides the model performance, the posterior parameter distribution may also give some insight about the value of RS-ET data in model calibration. The present study highlights the effectiveness of incorporating RS-ET data in the calibration of the SWAT model, which can lead to improve the simulation accuracy of runoff, while also ensuring the precision of ET simulation, ultimately resulting in more accurate hydrological modeling results. Even GLUE treats a parameter set as a whole to evaluate the possibility of the being an acceptable simulator of the system, the response of an individual parameter to the calibration data is valuable to assess the sensitivity of the model to that parameter. The results show that for the Experiment I, the parameters that posterior distribution takes on strong difference with assumed uniform prior distribution are mainly related to lateral flow and runoff generation, which mostly do not overlap with the ones identified by Experiment II, which mainly related to snow-melting, soil water movement and evaporation processes.

The differences between the two experiments indicate that the streamflow and RS-ET data differ in constrain feasible parameter space. What is worth-noticing is the results of Experiment III, the identified parameters include all the parameters being detected in Experiment I and II, and two additional parameters related to surface runoff generation. A common phenomenon of hydrological modeling is that various parameter sets yield similar model outcome, i.e., equifinality. More parameters are identified as being sensitive indicates that incorporating RS-ET into streamflow data for model calibration are





effective in reducing the degree of equifinality and further confirmed the effectiveness of applying such model calibration strategies in the QTP.

## 4.2 Implications for hydrological modelling in QTP

Although the posterior distribution of parameters identified by GLUE partially reflects parameter sensitivity, under the GLUE framework, it is difficult to quantitatively evaluate the impact of each parameter or interaction between parameters on model simulation, which are important for guiding the selection of the parameter need to be calibrated and improving the understanding about the characteristics of the water cycle in the basin. Therefore, the Sobol' method was applied to gain a whole picture about parameter sensitivity and interactions. For the three experiments, the number of parameters being regarded

as being sensitive according to ST are almost same (five, five and six), and is much lower than the original number of parameter (twenty-eight) being analyzed. This is consistent with the results of Zhang et al. (2013) that only a few parameters in the SWAT model have significant influences on model output. These finding are useful to reduce the number of parameters need to be calibrated and subsequently improve the efficiency of automatic calibration. Although the number of sensitive parameters is almost the same for the three experiments, most parameters are related to different hydrological processes. For Experiment III,

the identified six parameters characterize the surface runoff generation, snow-melting, soil water movement and evaporation processes, and had been recognized sensitive in Experiment I or II. It is found that the sum of ST for the top three parameters in Experiment III (0.53) is lower than that of Experiment I and II (0.73 and 0.63) and consequently implied that more parameters influence model output. Meanwhile, the contribution of parameter interaction to the total variance are highest in Experiment III. By these facts, it is indicated that RS-ET data bring more information about the hydrological cycle into the

model calibration and more modules in the model are activated, which also partly explains that more measured data are encompassed in the simulated uncertainty band, which becomes wider than calibration solely based on streamflow data.

It is also noticing that, the S1 is lower than ST for most of sensitive parameters and there are detected pairs of parameters that their interactions make contributions to variance of model output, more specifically, parameters related to soil property and snow-melting processes. However, the Latin hypercube sampling under GLUE doesn't explicitly consider joint

distribution of model parameters with strong correlation during the random sampling processes. To further reducing simulation uncertainty raised from model parameter under the proposed calibration strategies, updating prior parameter distribution with new observation or information about the parameter is a potential approach. For parameters related to soil water movement, Sun et al. (2016) employed a pedotransfer function estimated three soil parameters (SOL_AWC, SOL_K, and SOL_BD) and effectively reduced the uncertainty of the SWAT simulation, and suggested that the reduction in uncertainty could be attributed

to a better representation of soil moisture characteristics compared with parameter values gained from calibration. In QTP, many studies related to cryosphere may provide valuable information for updating prior distributions of SWAT model parameters. For example, Liu et al. (2018) applied statistical methods to estimate ranges of critical temperature for precipitation phase separation in the QTP region, which corresponds to the parameter SFTMP in the SWAT model. Li et al. (2021) conducted a comprehensive observation work of water heat transfer and effective thermal conductivity of a snowpack lasted



more than five months in the Ngoring Lake basin of the QTP and collected many valuable information about timing and threshold of snow-melting, which are possibility useful for put a better constraint on the SWAT parameter SMTMP, denoting the critical air temperature at which substantial snowmelt occurs.

## 5 Conclusion

Reducing simulation uncertainty is always an important issue of hydrological modeling. For the QTP, where the hydrological
processes are unique due to high altitude and cold weather, the lack of in-situ observation data brings great challenges for rainfall-runoff modeling. This study attempted to make a thorough investigation how incorporating RS-ET data into calibration could improve hydrological modelling in QTP, through the case of Yalong River basin using SWAT model. Three calibrations using streamflow data, RS-ET, and a combination of streamflow and RS-ET data were carried out respectively. The results show that when the SWAT model was calibrated with streamflow or RS-ET solely, the performance of the simulated variable
for which the data was applied for calibration is satisfactory, while the accuracy of simulation for the other variable is low. Compared with calibration using streamflow data solely, combing both the RS-ET and streamflow data for model calibration could improve the simulation accuracy for the two hydrological variables and uncertainty band of simulation could embrace more observations. Meanwhile, calibration using both types of observations could increase the number of parameters that posterior distributions are different from assumed uniform prior distribution, which indicate the degree of equifinality was
reduced. A more comprehensive parameter sensitivity analysis by the Sobol' method revealed that no more than six parameters out of the 28 parameters are adequate to account for model output variability. However, the detected sensitive parameters, their rankings and significant pairwise interactions differ among the three experiments, which explains the difference of model performances among the three experiments. The sensitive parameter detected based on both types of observations covers surface runoff generation, snow-melting, soil water movement and evaporation processes, while using single type of
observations, the identified sensitive parameters are only the ones related the hydrological processed quantified by the observations. It is indicated from these findings that by integrating RS-ET data into streamflow data for SWAT model calibration, not only the model output performs better, but also the characteristics of water cycle are captured by the calibrated model more effectively, highlighting the necessity of incorporating RS-ET data for hydrological model calibration in QTP. To further reduce simulation uncertainty after applying the calibration strategy proposed in this study, updating prior distribution
of parameter related to soil property or snow-melting processes when new data or information becomes available is a promising approach.

## Code and data availability

The DEM dataset can be downloaded from Geospatial Data Cloud: https://www.gscloud.cn/. The Land Use and Land Cover (LULC) dataset can be obtained from National Earth System Science Data Center, National Science & Technology
Infrastructure of China: http://www.geodata.cn/. The soil data is available at Institute of Soil Science, Chinese Academy of Sciences: http://vdb3.soil.csdb.cn/.The gridded precipitation dataset Multi-Source Weighted Ensemble Precipitation



(MSWEP) V2.8, which was developed by Beck et al. (2019), and can be downloaded from: http://www.gloh2o.org/mswep/. The gridded temperature dataset obtained from Multi-Source Weather (MSWX) developed by Beck et al. (2022), and can be downloaded from: http://www.gloh2o.org/mswx/. The Global Land Evaporation Amsterdam Model (GLEAM) developed by researchers Miralles et al. (2011) from the Department of Hydrology and Meteorology, School of Geographic Sciences at the University of Bristol, UK, and can be downloaded from: https://www.gleam.eu/. The code for model calibration and streamflow data are available from the corresponding author (sunny@bnu.edu.cn) upon request.

## Author contribution

JW and WS developed the methodology. JW developed the code, did the simulation and analysis. JW, LZ and WS wrote the original manuscript. JW and CM processed the data and visualized the results.

## Financial support

This study was supported by the National Key Research and Development Program of China (Grant no. 2022YFC3204403), National Natural Science Foundation of China (Grant no. 52179002), and the 111 Project (B18006).

## Competing interests

The contact author has declared that none of the authors has any competing interests.

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
