# Peer review of "How Remote-Sensing Evapotranspiration Data Improve Hydrological Model Calibration in a Typical Basin of Qinghai-Tibetan Plateau Region"

_Hydrology and Earth System Sciences, 2023_

## Referee Comment (RC2)

Dear editors,

Included here is my review of the manuscript submitted to the *Hydrology and Earth System Sciences*.

Manuscript #: hess-2023-200
Title: How Remote-Sensing Evapotranspiration Data Improve Hydrological Model Calibration in a Typical Basin of Qinghai-Tibetan Plateau Region

The hydrology processes in the Qinghai-Tibet Plateau are important, and the hydrological models are useful tools for simulating these processes. In this work, the authors combined stream flow data and the RS-ET data from GLEAM in the calibration to improve the accuracy of SWAT model in Yalong River Basin of the Qinghai-Tibet Plateau. The authors did a large amount of calculations and the data analysis is solid and convincing. However, in my opinion, the findings in this manuscript are not beyond the general understanding, thus, provides few new knowledge to the science public. I feel sorry, but I need to reject this manuscript to keep the high quality of this journal.

The detail comments and suggestions are listed as following:

**General comments:**
 1. Evaporation is an important component in the water balance. It is not surprising that adding another calibration target of evaporation can improve the accuracy of the hydrological models. In the introduction section, the authors cited some previous studies (e.g., Immerzeel and Droogers (2008); Huang et al. (2020)) which have already evaluated the efficiency of such improvement in India, the Yalong River basin, and other basins. What's the advance of this study compared with those previous studies? This model (SWAT in Immerzeel and Droogers (2008)) and this region (the Yalong River basin in Huang et al. (2020)) have already been investigated. Someone can write dozens of papers by combining different hydrological models and different RS-ET datasets applied in different hot zones about this topic, which has fewer contributions to the academic society. Thus, I did not see the innovations of this study. I strongly recommend the authors change the primary purpose of this manuscript. Elaborating the story from another perspective using the calculation they already did will be a wise choice.

2. I did not see any correction about the evaporation data from GLEAM 3.5a. The authors seemed to extract the grid data from the GLEAM dataset directly without any local bias correction, which is not proper. I suggest that the authors calibrate and correct the evaporation data from GLEAM first, since Huang et al. (2020) have already evaluated the different performance in the hydrological models between bias-corrected and nonbias-corrected evaporation data.

3. The authors spent a lot of effort on the sensitive analysis. I really respect the extensive work and understand that they tried to provide understandings of the driving mechanism, which has the potential to be regarded as innovations. However, this manuscript only listed the statistical results with a few simple discussions, which cannot elaborate on the driving mechanism. The authors should offer additional deep analysis of these sensitive results with physical meanings.

**Other Specific comments:**
 1. The abstract is too long, and the authors can simplify it for the convenience of the reader.

2. Line 35, the improvement of NSE (e.g., from 0.71 to 0.81) is not significant compared with the previous studies (e.g., from 0.41 to 0.81 in Immerzeel and Droogers (2008))

3. Lines 56-56, any connection with this study's topic?

4. Lines 276-277, why?

5. Figure 4b, what's the reason for the underestimation of Q between 2004 and 2005

6. In Figure 8, the authors can try only to provide the key parameters in the figure to reduce reader interference.

7. Line 346, "when combing streamflow and RS-ET data for model calibration, the accuracy of simulated streamflow and ET are all higher." I remember that the accuracy of evaporation in experiment two is higher than in experiment three (Lines 276-277).

---

## Author Comment (AC1)

Dear reviewer,

Thank you very much for your constructive comments and suggestions. In the texts below, we will try to answer all questions addressed by the reviewer. If you feel more explanations or revisions are needed. Please do not hesitate to contact with us.

Best regards,

From the authors

**This is an interesting and timely study. Reducing simulation uncertainty has always been an important issue of hydrological modeling. Traditionally, hydrological models are calibrated and validated using only runoff data, which not only leads to parameter equivocality and failure to obtain reasonable and true parameters, but also leads to large uncertainties in other elements of the simulation such as evapotranspiration, soil water. This study attempts to explore how incorporating RS-ET data into the calibration could improve hydrological modelling. I think there is potential in this manuscript, but there are several sections that need to be more clearly explained My comments are as follows,**

**1. There are a large variety of ET products, why choose GLEAM data and how accurate are GLEAM data in this basin? A set of ET model data may have large uncertainties both in magnitude and in spatial and temporal distribution, and direct use without calibration may introduce greater uncertainty. It is recommended that a water balance analysis, which analyzes relationships between the precipitation, runoff, and ET data used in the study, be added to determine the overall confidence in the data and thus improve the credibility of the article's results.**

**Response:**
Thank you very much for this suggestion. To evaluate the accuracy of GLEAM ET data, a water balance at the studied basin was conducted at annual scale. For the upstream area of the Ganzi Gausing station, which is the outlet of hydrological modelling in this study, the runoff ($Q_{est,1}$) was estimated from the area averaged precipitation (derived from MSWEP dataset as input for SWAT model; the analysis about accuracy of MSWEP data is described in the responses to the following comment) minus GLEAM ET and compared with observed value at the Ganzi Station ($Q_{obs}$). Figure R1 is the scatterplot of $Q_{obs}$ and $Q_{est,1}$. Absolute bias (ABIAS) was computed to evaluate the accuracy:

$$\text{ABIAS} = \frac{\sum_{i=1}^{n}|Q_{obs,i} - P_{sim,i}|}{\sum_{i=1}^{n} G_{obs,i}} \tag{1}$$

where $Q_{obs,i}$ and $Q_{sim,i}$ represents observed runoff and estimated runoff from water balance analysis at time step $i$, respectively, $n$ is the number of samples. The ABIAS for $Q_{est,1}$ is 0.20.

Huang et al. (2020) used bias-corrected PML-AET data to calibrate Xinanjiang hydrological model. Compared with the accuracy evaluation of PML-AET in the whole Yalong River Basin (Huang et al., 2020), the ABIAS of GLEAM ET is much lower than the uncorrected PML-AET (ABIAS: 0.55). As a high-attitude region, the snow-melting process has influences on runoff in the simulated basin of this study. We further corrected $Q_{est,1}$ by subtracting annual snow-melting amount (derived from Monthly Snowmelt Dataset in China during 1951-2020 (Yang et.al, 2022)) from $Q_{est,1}$ (mentioned as $Q_{est,2}$) and compared with $Q_{obs}$ (Figure R1). The ABIAS of $Q_{est,2}$ is 0.12, which is also lower that the bias-corrected PML-AET data (ABIAS: 0.18, Huang, et al., 2020). It is indicated that partly the error of $Q_{est,1}$ comes from ignoring the snow-melting process and cannot contribute to the GLEAM ET data completely. Based on literature review, the GLEAM ET has been wide used worldwide for analyzing changes in regional water cycle (e.g., Bennour, et al., 2022, Ding and Zhu, 2022), rainfall-runoff modelling (e.g. Dembélé, et al., 2020, López López, et al., 2017). Also, from the results of model calibration in this study, incorporating GLEAM ET data into SWAT model calibration did improve the accuracy of runoff simulation. Based on these facts, we are confident using the current version of GLEAM ET data in this study.

The above-mentioned analysis will be added to the results section of the manuscript.

[Figure]

**Figure R1.** Observed annual runoff versus the runoff estimated from the area averaged precipitation (MSWEP) minus GLEAM ET ($Q_{est,1}$), and versus the ones obtained from precipitation minus GLEAM ET and annual snow-melting ($Q_{est,2}$)

**1.Similarly, is it possible to validate or document the regional applicability of climate-driven data?**

**Response:**

Precipitation data from two meteorological stations operated by China Meteorological Administration within the upstream region of Ganzi gauging station are available to evaluate the MSWEP precipitation data the we used to drive the hydrological model. The scatterplots for daily, monthly and annual precipitation between ground and satellite precipitation data at

the pixels corresponding to the two meteorological stations (2001 to 2010) are shown in Figure R2. The correlation coefficient for daily, monthly and annual is 0.52, 0.94 and 0.85, respectively. The accuracy is highest for the monthly scale, which is the temporal scale for the hydrological model simulation.

The above-mentioned analysis will be added to the results section of the manuscript.

[Figure]

**Figure R2.** Comparison between observed precipitation and MSWEP precipitation at (a) daily, (b)monthly and (annual) scale

**2.In section 2.1, what are the values for the percentage of runoff sources roughly, this could be crucial information. It is also recommended that the percentage of area of major soil types and LULC types be given. Although this may have been shown in the figure, it would be easier for the reader to understand if specific values were given.**

**Response:**
Based on the analysis of MSWEP precipitation data and streamflow data for the period of 2001 to 2010, the rainfall-runoff ratio is 0.59. The percentage of percentage of area of major soil types has been shown in the following table (Table R1). The major land use and land cover type are grass land, forest and bare land, which occupies 78.9%, 13.2% and 12.4% of the basin area, respectively.

These information will be added to Section 2.1 Study area.

**Table R1** The percentage of area of major soil types

| Soil Type | Grey Brown Soil | Swamp Soil | Grass Felt Soil | Thin Grass Felt Soil | Brown Grass Felt Soil | Black Felt Soil | Permafrost Soil |
|---|---|---|---|---|---|---|---|
| Percentage of Area | 3.36% | 2.64% | 56.23% | 7.69% | 4.43% | 15.78% | 9.87% |

**3.In Section 3.1, it is desired to make a multidimensional comparison of the analysis of information in figures and tables, e.g., a comparison of NSEQ and NSEET in the same experiment.**

**Response:**
Thank you very much for the comment. The accuracy and uncertainty of streamflow and ET

simulation will be compared in the revised paper through the information provided by Table R2.

For Experiment I using streamflow data solely for calibration, the $NSE_{50\%}$ for streamflow simulation in the calibration and validation period is much higher than those of ET, which is all lower than 0 and means that the performance of ET simulation is unsatisfactory. The P_factor corresponding to the streamflow estimation is higher than that for ET simulation, which means more observation are embraced by the uncertainty band. Meanwhile, The R_factor for streamflow estimation is lower than ET, indicating the width of uncertainty band is narrow. For modeling accuracy and simulation uncertainty, streamflow estimation all outperforms ET simulation. For Experiment II using RS-ET data solely for calibration, the $NSE_{50\%}$ for ET simulation in the calibration and validation period is higher than those of streamflow. The P_factor corresponding to the ET estimation is similar with the value for streamflow simulation. The R_factor for streamflow estimation is higher than ET, implying uncertainty of streamflow estimation is higher than ET simulation. For Experiment III combing both of streamflow and ET data for model calibration, the $NSE_{50\%}$ for streamflow simulation is slightly higher than that for ET simulation. P_factor is in the same level for the simulation of the two hydrological variables. The R_fator of ET is a little bit higher than that for streamflow simulation. All these factors demonstrated that the streamflow simulation performs better to a small degree.

**Table R2.** Accuracy and uncertainty for streamflow and ET simulations

| | | Experiment I | | Experiment II | | Experiment III | |
|---|---|---|---|---|---|---|---|
| | | Calibration | Validation | Calibration | Validation | Calibration | Validation |
| $NSE_{50\%}$ | Q | 0.71 | 0.75 | 0.52 | 0.54 | 0.81 | 0.84 |
| | ET | -0.22 | -0.09 | 0.84 | 0.88 | 0.79 | 0.79 |
| P_factor | Q | 0.75 | 0.75 | 0.88 | 0.88 | 0.85 | 0.87 |
| | ET | 0.58 | 0.52 | 0.93 | 0.88 | 0.88 | 0.88 |
| R_factor | Q | 0.86 | 0.92 | 1.42 | 1.51 | 1.1 | 1.17 |
| | ET | 1.42 | 1.35 | 1.2 | 1.14 | 1.18 | 1.18 |

**4.In Section 3.2, why is the number of behavioral parameter sets different in the three experiments? Is it because the number of parameters sensitive to evapotranspiration processes in a hydrological model like SWAT is much smaller than the number of parameters sensitive to runoff processes?**

**Response:**

We agree with the reviewer that, a complex model like SWAT, the number of parameters related to evapotranspiration processes is lower than the ones connected with runoff processes, because as the integrated output of water cycle at basin scale, the runoff is determined by many hydrological processes with a basin, which also include evapotranspiration processes. For the three experiments, the data used for calibration is different, which is observed streamflow data, remote sensing evapotranspiration, and the combination of the streamflow and evapotranspiration data. The automatic calibration process tried to minimize the difference between observation data and model simulation by searching the parameter space. As the

calibration data are different among the three experiments, the parameter sets being gained by calibration are also different, which explains why is the number of behavioral parameter sets is different.

**5.The headings of sections 3.1 and 3.2 are the same.**

**Response:**
The heading of section 3.2 will be changed into "Comparison of parameter posterior distributions among three experiments"

**6.There should be an error in the x-axis in set (b) of Figures 2-7.**

**Response:**
The years showed in the x-axis of Figures 2-7 will be corrected.

**7.Figures 2-7 could be merged into one or two figures. And Tables 3-4 could be merged into one table. In Figure 6, "b" was mislabeled as "d".**

**Response:**
The figure 2 to 4 are merged in to one figure. Similarly, the Figure 5 to 7 are merged into one figure. The merged figures are shown as Figure R3 and R4. Tables 3 and 4 are merged into one table as Table R2. The sequence number of each figure has been corrected.

[Figure]

**Figure R3.** Streamflow observation, best simulation (50% quantile of ensemble simulation) and uncertainty band for the calibration and validation period corresponding to Experiment I (a) and (b), Experiment II (c) and (d), and Experiment III (e) and (f)

[Figure]

**Figure R4.** Evapotranspiration observation, best simulation (50% quantile of ensemble simulation) and uncertainty band for the calibration and validation period corresponding to Experiment I (a) and (b), Experiment II (c) and (d), and Experiment III (e) and (f)

**8.Figure 12 is missing (d) in the title**

**Response:**

This typo will be corrected in the revised manuscript.

**References:**

Bennour, A., Jia, L., Menenti, M., Zheng, C., Zeng, Y., Asenso Barnieh, B., Jiang, M.: Calibration and Validation of SWAT Model by Using Hydrological Remote Sensing Observables in the Lake Chad Basin. Remote Sensing, 14(6): 1511. doi:10.3390/rs14061511, 2022.

Dembélé, M., Hrachowitz, M., Savenije, H.H.G., Mariéthoz, G., Schaefli, B.: Improving the Predictive Skill of a Distributed Hydrological Model by Calibration on Spatial Patterns With Multiple Satellite Data Sets, Water Resources Research, 56(1), doi:10.1029/2019WR026085, 2020b.

Ding, J., Zhu, Q.: The accuracy of multisource evapotranspiration products and their applicability in streamflow simulation over a large catchment of Southern China. Journal of Hydrology: Regional Studies, 41: 101092. doi: 10.1016/j.ejrh.2022.101092, 2022.

Huang, Q., Qin, G., Zhang, Y., Tang, Q., Liu, C., Xia, J., Chiew, F.H.S., Post, D.: Using Remote Sensing Data-Based Hydrological Model Calibrations for Predicting Runoff in Ungauged

or Poorly Gauged Catchments, Water Resources Research, 56(8), doi:10.1029/2020 wr028205, 2020.

López López, P., Sutanudjaja, E.H., Schellekens, J., Sterk, G., Bierkens, M.F.P.: Calibration of a large-scale hydrological model using satellite-based soil moisture and evapotranspiration products, Hydrology and Earth System Sciences, 21(6), 3125-3144, doi:10.5194/hess-21-3125-2017, 2017.

Yang, Y., Chen, R., Liu, G., Liu, Z., Wang, X.: Trends and variability in snowmelt in China under climate change, Hydrology and Earth System Sciences, 26: 305-329. doi: 10.5194/hess-26-305-2022, 2022.

---

## Author Comment (AC2)

**Responses to reviewer #2**

Dear reviewer,

Thank you very much for your constructive comments and suggestions. In the texts below, we will try to answer all questions addressed by the reviewer. If you feel more explanations or revisions are needed. Please do not hesitate to contact with us.

Best regards,

From the authors

**The hydrology processes in the Qinghai-Tibet Plateau are important, and the hydrological models are useful tools for simulating these processes. In this work, the authors combined stream flow data and the RS-ET data from GLEAM in the calibration to improve the accuracy of SWAT model in Yalong River Basin of the Qinghai-Tibet Plateau. The authors did a large amount of calculations and the data analysis is solid and convincing. However, in my opinion, the findings in this manuscript are not beyond the general understanding, thus, provides few new knowledge to the science public. I feel sorry, but I need to reject this manuscript to keep the high quality of this journal.**

**Response:**

Thank you very much for this important comment. We apologize that the originality of this study was not described clearly. In the revised manuscript, we will clarify the differences between our study and previous ones and the scientific question that we are targeting:

Calibrating hydrological model using RS-ET data has been studied by many researchers to solve the problem of lack streamflow data for model calibration. However, in real world, the degree of lacing streamflow data varies significantly among different basins: There are basins without any ground streamflow observation; there are also basins that streamflow observations are available at basin outlet but no observations are available inside the basin. The calibration problem of hydrological model is a board scientific question, based on the availability of ground and satellite data, the strategies of dealing with the calibration problem could be quite different. Previous studies focus on using RS-ET data in totally ungauged basins (A detailed summary of literature review is available in the responses to general comment one in Table R1). There are also studies combing streamflow and RS-ET data for model calibrations and comparing the model performance among different calibration strategies. It is found that whether using RS-ET data for model calibration or not will lead to differences in model output. However, the reasons contribute to such differences have not been explored completely, which is an important issue to gain insights about simulation uncertainty and correspondingly to build confidences applying the model calibrated based on RS-ET data for real-world application.

In the remote and high-altitude basins of Qinghai-Tibet Plateau, ground observations of streamflow are only available in limited sites. When calibrating distributed hydrological model using only streamflow data in basin outlet, the simulation uncertainty may be high, due to the complex model structure and high dimensional parameter space. Combing RS-ET data with streamflow data for model calibrating may further reduce simulation uncertainty compared with

using streamflow data alone. This is the scientific question that this study tries to address. To make one more step further than previous studies, the mechanisms of leading to the differences among model behavior when using different calibration data are evaluated through parameter sensitivity analysis, which is an important approach to judge whether the calibrated model reflect the unique characteristics of the water cycle in the studied basin. In our opinion, the findings from this study could provide a thorough understanding about the values and limitations of using RS-ET for hydrological model calibration in the Qinghai-Tibet Plateau and provide guidance for further reducing simulation uncertainty in this cold and high-altitude region. The knowledge gained in this study will draw board attention from scientific communities of hydrological modelling and remote sensing application in hydrology.

**The detail comments and suggestions are listed as following:**
**General comments:**
**1. Evaporation is an important component in the water balance. It is not surprising that adding another calibration target of evaporation can improve the accuracy of the hydrological models. In the introduction section, the authors cited some previous studies (e.g., Immerzeel and Droogers (2008); Huang et al. (2020)) which have already evaluated the efficiency of such improvement in India, the Yalong River basin, and other basins. What's the advance of this study compared with those previous studies? This model (SWAT in Immerzeel and Droogers (2008)) and this region (the Yalong River basin in Huang et al. (2020)) have already been investigated. Someone can write dozens of papers by combining different hydrological models and different RS-ET datasets applied in different hot zones about this topic, which has fewer contributions to the academic society. Thus, I did not see the innovations of this study. I strongly recommend the authors change the primary purpose of this manuscript. Elaborating the story from another perspective using the calculation they already did will be a wise choice.**
**Responses:**
Thank you very much for helping us to clarify the contributions of this study. We agree with the reviewer that there are many studies that have already touch the topic of calibrating hydrological model using RS-ET data. In order to show the progresses and limitations of these studies and subsequently clarify the originality of our study, we made an intensive literature review and summarized their founding in Table R1. The degree of model structure complexity in these studies varies significantly, ranging from lumped conceptual models (e.g., Vervoort et al., 2013; López et al., 2017) to distributed physically-based distributed model (e.g., Immerzeel and Droogers, 2008; Herman et al., 2018). All these studies used RS-ET data alone or combing it with other type of observations for hydrological model calibration and tried to evaluate the usefulness of RE-ET data from the aspect that whether the model performance has been improved. However, after using RS-ET data, whether model behavior is more consistent with current hydrologic understanding of reality has not been discussed sufficiently, which is an important issue to build confidence for applying model to solve real world problem. In this context, parameter sensitivity analysis was recommended to investigate model behavior (Dembélé et al., 2019). Moreover, Moazenzadeh and Izady (2022) considered that knowledge about parameter interactions is valuable to understand the effects of the specific variables on hydrological components of water cycle at basin scale. But these issues related to parameter uncertainty has not been examined intensively. In this study, we tried to answer this question

by a comprehensive analysis of parameter response surface conditioned on different calibration data using Sobol sensitivity analysis method, which can quantity influence sensitivity of single parameter and parameter interaction, and subsequently explore the mechanisms leading to the improvement of model performance after calibrating using RS-ET data.

We would like to explain the difference between our study and two paper mentioned by the reviewer. The study of Immerzeel and Droogers (2008) is among the pioneer work of calibration using RS-ET data. They used RS-ET data solely and evaluated the calibrated model by the fitness of reproducing monthly ET and historical streamflow. In our study, we combined RE-ET with streamflow observation for model calibration and tried to show the improvement of model performance compared with calibration using streamflow data solely and explains the possible mechanism of such improvement. The Huang et al. (2020) also conducted the study in the whole Yalong Basin. Considering the great differences in climate (varies from continental plateau climate to subtropical humid climate) and altitude (varies greatly from 6,000 m to 1000 m), they divided the whole basins into 30 catchments and calibrate the model in each catchment using RE-ET data and compared model output with the ones gained from traditional regionalization approach. Our study focuses on model calibration in the most upstream catchment of Huang et al. (2020), the upstream region of Ganzi Gauging station, which is located in the Qinghai-Tibet Plateau region and accounts for one fourth of the entire area of Yalong Basin. Huang et al. (2020) centered upon the advantages of using bias-corrected RE-ET data for model calibration in 30 catchments with wide ranges of natural condition, while our study pay attention to high-mountain regions and conducted more detailed analysis about how RE-ET data could constrains model behavior from the aspect of parameter model response surface and its influences on model simulation of snow-melting processes, which is also mentioned by Huang et al. (2020) that should be considered in the future hydrological modelling works in the Yalong basin. In our opinion, the two studies mentioned by the reviewer target at totally ungagged basin, while our study target at how RS-ET data could further improve model performance in the case that streamflow data at basin outlet are available for model calibration.

After carefully consideration, we will revise the storyline of this study as follows: In the high-altitude and data-sparse Qinghai-Tibet Plateau, streamflow data at basin outlet may be available for distributed hydrological model calibration. However, interactions among multiple internal processes of water cycle in this unique region cannot be captured by streamflow data alone, which will bring uncertainty to model simulation. In such context, incorporating RS-ET with streamflow data for model calibration may reduce uncertainty in simulation of internal hydrological process and finally improve the streamflow prediction at basin outlet. The main objective of this study is to verify this hypothesis in a typical basin of the Qinghai-Tibet Plateau, i.e., the upstream catchment of Ganzi gauging station. The value of RE-ET data in this context will be evaluated from the aspects of: 1) whether model performance is improved after calibration, .i.e., the accuracy and uncertainty of streamflow and ET simulation; 2) whether model behavior has reflected the reality of hydrological characteristics, via examining the relationship between model performance and model behavior through a physical interpretation about detected sensitivity of single parameter and parameter interaction. The second aspect has not been examined thoroughly in previous studies and could elaborate the value of RE-ET data with more physical meanings.

**Table R1.** Summary of studies in which RE-ET data were used to calibrate a hydrological model. N/D: "not provided"; N/A: "not applicable". In the last column the numbers refer to: (1) The performance of the RS-ET data for model calibration;(2)Identified parameter sensitivity; (3) The mechanism that leads to the difference of model performance in calibration between using RS-ET and other type of observations

| Study | RS-ET data being used | Correction for RS-ET data | Location/Basin Size /Annual precipitation | Hydrological model /calibration method | Key calibration schemes compared | Key findings |
|---|---|---|---|---|---|---|
| Immerzeel, 2008 | MODIS ET (SEBAL method) | No | Upper Bhima Basin/ 45,678 km$^2$/941 mm | Distributed SWAT model/ Gauss–Marquardt–Levenberg Gradient Search method | RS-ET data | 1. Accuracy of monthly sub-basin simulated ET improved. 2. ET was more sensitive to the groundwater and meteorological parameters than the soil and land use parameters 3.N/A |
| Rientjes et al. (2013) | MODIS-ERRA ET (SEBS method) | No | 7 subbasins in Karkheh River Basin, Iran/ 1285-9873km$^2$/150-750 mm | Lumped conceptual HBV model/ Monte Carlo method | 1.Streamflow 2.RS-ET 3. Streamflow + RS-ET | 1. Catchment water balance is best reproduced when both Streamflow and RS-ET serve as calibration target 2. N/A 3.Parameters have specific optimum values depending on the selected calibration variable. |
| Willem Vervoort et al. (2014) | MODIS ET (16A3) | No | 4 subbasins located in Murrumbidgee Basin, Australia/146.5~2183 km$^2$/600-1100 mm | Lumped IHACRES model/SCE-UA | 1.Streamflow 2.RS-ET 3. Streamflow + RS-ET | 1.For the conceptual models used in this study, RS-ET did not improve the calibration results 2. N/A 3.Limited number of model parameters constrains degrees of freedom for the model to adjust to the new data |
| López et al., 2017 | GLEAM ET | No | Oum er Rbia River, Morocco/38025 km2/400 mm | Lumped conceptual PCR-GLOBWB Model | 1.Streamflow 2.RS-ET 3.CCI soil moisture (SM) 4. RS-ET+CCI SM | 1. A model calibrated only on RE-ET or CCI soil moisture data achieved a lower discharge performance than when they used together. 2.N/A 3.Calibration using only GLEAM ET or only CCI soil moisture data can result in over-parametrization or equifinality problem. |
| Herman et al., 2018 | MODIS ET (SSEBop method), GOES ET (ALEXI method) | No | Honeyoey Creek-Pine Creek Watershed/ 1100 km$^2$/N/D | Distributed SWAT/Monte Carlo method, NSGA-II | 1.Streamflow 2. Streamflow + RS-ET | 1. Both calibration schemes can improve the ET simulation using RS-ET data but their influences on streamflow simulation differs. 2. N/A 3. N/A |
| Dembéléet al., 2019 | GLEAM ET | No | Volta River basin in West Africa/415600 km$^2$/470-1420 mm | Distributed mHMm model/ dynamically dimensioned search algorithm | 1.Streamflow 2.Three satellite products of ET, soil moisture and terrestrial water storage (GRACE) | 1.Worse performance of multivariate calibration for streamflow and terrestrial water storage is counterbalanced with an increase in performance for soil moisture and evaporation. 2. N/A 3. N/A |

| Study | RS-ET data being used | Correction for RS-ET data | Location/Basin Size /Annual precipitation | Hydrological model /calibration method | Key calibration schemes compared | Key findings |
|---|---|---|---|---|---|---|
| Zhang et al, 2020 | PML ET | No | 222 catchments across Australia/ N/D | Lumped Xinanjiang and SIMHYD models/ A genetic algorithm | 1.Streamflow
2.RS-ET
3.RS-ET+ Fu model estimated mean annual runoff | 1. Performance from RE-ET calibration are encouraging, particularly in monthly runoff and mean annual runoff in the wetter catchments
2. N/A
3. N/A |
| Huang et al., 2020 | PML ET | Yes | 30 subbasins in the Yalong River Basin/720 mm | Lumped Xinanjiang and model/ A genetic algorithm | 1.Streamflow
2.Raw RS-ET
3.Corrected RS-ET | 1. Using bias-corrected RS-ET data to calibrating hydrological models has great potential to estimate daily and monthly runoff time series.
2.N/A
3. It is hard to find general rules between performance metrics and catchment attributes with merely 30 sampling basins |
| Liu et al., 2022 | GLEAM ET, PML ET and MODIS ET (16A) | Yes | 59 large basins Worldwide/157-2261 mm | Lumped conceptual abcd and the DWBM model/NSGA-II | 1.Streamflow
2. Corrected RS-ET
3. Corrected RS-ET and terrestrial water storage (GRACE) | 1. Calibration by combining corrected RS-ET and terrestrial water storage can simulate runoff well, but worse than using streamflow.
2. N/A
3. Runoff simulation accuracy based on RS-ET significantly decreased with the increase in aridity |
| Moazenzadeh and Izady, 2022 | MODIS ET (SEBAL method) | No | 3 basins of Neishaboor watershed, Iran/ 107~9158 km$^2$/261 mm | Distributed SWAT model/SUFI2 | 1.Streamflow
2.Hybrid calibration (Firstly streamflow; then RS-ET) | 1. The hybrid calibration method significantly improved the accuracy of the streamflow estimation.
2. Relative sensitivities based on linear approximations were detected.
3. Using RS-ET against streamflow data in hybrid calibration helps to consider parameter interactions more accurately. |
| This study | GLEAM ET | No | Upstream Basin of Ganzi Gausing station in Yalong River, China/32535 km$^2$/530 mm | Distributed SWAT model/GLUE | 1.Streamflow
2.RS-ET
3.Streamflow+RS-ET | 1. Simulation accuracy and uncertainty of Streamflow and ET are compared among the three calibration strategies.
2. Sensitivity of single parameter and parameter interaction are evaluated quantitively using Sobol method.
3. The relationship between model performance and model behavior will be examined through the physical interpretation of sensitivity of single parameter and parameter interaction. |

**2. I did not see any correction about the evaporation data from GLEAM 3.5a. The authors seemed to extract the grid data from the GLEAM dataset directly without any local bias correction, which is not proper. I suggest that the authors calibrate and correct the evaporation data from GLEAM first, since Huang et al. (2020) have already evaluated the different performance in the hydrological models between bias-corrected and nonbias-corrected evaporation data.**

**Responses:**

We totally agree with the reviewer that the accuracy of GLEAM ET data need to be evaluated before being applied for model calibration. We followed the method of Huang et al. (2020) to check the accuracy using a water balance analysis. For the upstream area of the Ganzi Gausing station, which is the outlet of hydrological modelling in this study, the runoff ($Q_{est,1}$) was estimated from the area averaged precipitation (derived from MSWEP dataset as input for SWAT model) minus GLEAM ET and compared with observed value at the Ganzi Station ($Q_{obs}$). Figure R1 is the scatterplot of $Q_{obs}$ and $Q_{est,1}$. The Absolute Bias (ABIAS) was computed to evaluate the accuracy:

$$\text{ABIAS} = \frac{\sum_{i=1}^{n}|Q_{obs,i}-P_{sim,i}|}{\sum_{i=1}^{n} G_{obs,i}} \tag{1}$$

where $Q_{obs,i}$ and $Q_{sim,i}$ represent observed runoff and estimated runoff from water balance analysis at time step $i$, respectively, $n$ is the number of samples. The ABIAS for $Q_{est,1}$ is 0.20. Huang et al. (2020) used bias-corrected PML-AET data to calibrate Xinanjiang hydrological model. Compared with the accuracy evaluation of PML-AET in the whole Yalong River Basin (Huang et al., 2020), the ABIAS of GLEAM ET is much lower than the uncorrected PML-AET (ABIAS: 0.55). As a high-attitude region, the snow-melting process has influences on runoff in the simulated basin of this study. We further corrected $Q_{est,1}$ by subtracting annual snow-melting amount (derived from Monthly Snowmelt Dataset in China during 1951-2020 (Yang et.al, 2022)) from $Q_{est,1}$ (mentioned as $Q_{est,2}$) and compared with $Q_{obs}$ (Figure R1). The ABIAS of $Q_{est,2}$ is 0.12, which is also lower that the bias-corrected PML-AET data (ABIAS: 0.18, Huang, et al., 2020). It is indicated that partially the error of $Q_{est,1}$ comes from ignoring the snow-melting process and cannot attribute the GLEAM ET data completely.

From the results of model calibration in this study, incorporating GLEAM ET data into SWAT model calibration did improve the accuracy of runoff simulation. Based on literature review shown in Table 1, most studies used RE-ET data directly for model calibration. One major objective of Huang et al. (2020) is to investigate difference in model performance of calibrations between bias-corrected RE-ET data and nonbias-corrected ones, while the objective of our study is different, which is to explore the mechanisms that driving the difference of model performances among three calibration strategies:1) using streamflow data solely, 2) using RS-ET data solely, 3) using streamflow and RS-ET data simultaneously. Based on these facts, we did not focus on bias-correction of GLEAM ET data. Instead, we will fully acknowledge that influences of error in RS-ET data on simulation uncertainty need to be considered in the discussion section of the revised manuscript.

[Figure]

**Figure R1.** Observed annual runoff versus the runoff estimated from the area averaged precipitation (MSWEP) minus GLEAM ET ($Q_{est,1}$), and versus the ones obtained from precipitation minus GLEAM ET and annual snow-melting ($Q_{est,2}$)

**3. The authors spent a lot of effort on the sensitive analysis. I really respect the extensive work and understand that they tried to provide understandings of the driving mechanism, which has the potential to be regarded as innovations. However, this manuscript only listed the statistical results with a few simple discussions, which cannot elaborate on the driving mechanism. The authors should offer additional deep analysis of these sensitive results with physical meanings.**

**Responses:**

Thank you very much for this constructive comment. To better elaborate on the driving mechanism of differences in the three calibrations, whether incorporating RS-ET data with streamflow data for model calibration will help the model more properly represent internal hydrological processes of water cycle at basin will be discussed. To this end, the physical meanings of identified sensitive parameters and their relationships with model behavior will be examined intensively. A new subsection will be added to the discission section: "Implications from the Sobol' sensitivity analysis". The content of the discussion is as follows:

Parameter sensitivity analysis can facilitate the understanding and interpretation of models. It has been considered as to be effective in distinguishing whether the parameters that control model response are representative for physical processes that dominate in reality (Pianosi et al, 2016). In this study, the model behaviors under different calibration strategies are examined through the sensitivity of single parameter and parameter interactions revealed by Sobol method. By doing this, the mechanism driving the difference in model output under different calibration strategies could be better explained.

For Experiment I using streamflow data solely for calibration, the ALPHA_BNK (Baseflow alpha factor for bank storage), LAT_TTIME (Lateral flow travel time), CN2 (SCS curve number), SMTMP (Snowmelt base temperature), and HRU_SLP (average slope steepness) are detected as sensitive parameters. The ALHPA_BNK characterizes the bank

storge recession in channel flow routing process. LAT_TTIME describe the lateral flow movement process. The CN2 determines the amount of surface runoff generated by certain amount of rainfall. These three parameters have direct influences on the streamflow volume at the basin outlet. HRU_SLP quantifies the influences of topography on surface flow and lateral flow, which also have close relationship with streamflow. Meanwhile, significant interactions between ALPHA_BNK and LAT_TTIME are found, which imply the fact that bank flow and lateral flow interaction also influence streamflow variations. SMTMP shapes the snow melting process, which is considered as a main component of runoff in the upper reach of Yalong River basin (Kang et al., 2001)

For Experiment II using RS-ET data solely for calibration, detected sensitive parameter include the SMTMP, SOL_AWC (Available water capacity of the soil layer), ESCO (Soil evaporation compensation coefficient), SOL_K (Saturated hydraulic conductivity of the soil layer) and HRU_SLP. ESCO directly affect the computation of soil water evaporation. SOL_AWC, SOL_K and HRU_SLP characterizes soil water vertical and lateral movement and consequently affect the amount of available soil water for evapotranspiration. Besides SMTMP itself, the interactions of two parameter related to cryosphere processes, SMTMP and SFTMP (Snow falling temperature) are also detected to be sensitive to the simulation of evapotranspiration. These results indicates that a close relationship between evapotranspiration and snow melting processes has been identified by the model conditioned on RS-ET data. This detected relationship is consistent with the finding of Guo et al. (2011) based on observations from the Coordinated Enhanced Observing Period/Asia-Australia Monsoon Project on the Tibetan Plateau.

Evapotranspiration at basin scale is one important hydrological process affect streamflow at basin outlet, which may explain that HRU_SLP and SMTMP are sensitive in both of Experiment I and II, as they all related to evapotranspiration and runoff generation processes. The other sensitive parameter and parameter interactions identified by in Experiment I describe hydrological processes regarding to runoff generation, but not associated much with evapotranspiration. This may lead to the fact that Experiment I have better performance in streamflow simulation but much worse estimations for evapotranspiration. As parameters characterizes evapotranspiration all relevant to runoff generation to some degree, the streamflow estimation in Experiment II performance worse than Experiment I, but not as worse as evapotranspiration simulation in Experiment I.

The differences in model performance and parameter sensitivity between Experiment I and II indicate that the information contained in streamflow and RS-ET data exert different constraints on model behavior. For Experiment III integrating these two types of data for model calibration, the identified sensitive parameters include the ones have been recognized sensitive in both of Experiment I and II (HRU_SLP and SMTMP), only in Experiment I (CN2), and only in Experiment II (SOL_AWC, SOL_K, ESCO). It is indicated that more model modules have been activated in Experiment III. Besides the interaction between SMTMP and SFTMP, the interaction between ESCO and TIMP (Snowpack temperature lag factor) which quantifies the influences of snowpack temperature of previous day on current day, are also significant in Experiment III, implying the interactions among evapotranspiration, soil water movement and snow-melting have been captured by the calibrated model. Meanwhile, the accuracy of streamflow and ET simulation all improved compared with Experiment I. Taking into account

the parameter sensitivity and model output in general, it is evident that the simulation behavior of the model in Experiment III can more comprehensively reflect the characteristics of the hydrological cycle in the basin than the model conditioned on streamflow data only.

**Other Specific comments:**
**1. The abstract is too long, and the authors can simplify it for the convenience of the reader.**

**Responses:**
The abstract has been rewritten and made more concise to convey key findings of this study to the readers. The revised abstract is as follow:

Distributed hydrological modelling provide valuable knowledge about water cycle and cryosphere of the Qinghai-Tibet Plateau (QTP). However, in this data-sparse region, models are usually calibrated based on streamflow data at basin outlet solely, which may not provide sufficient information for parameter estimations and correspondingly bring uncertainty to model simulation. The objective of this study is to thoroughly evaluate the value of incorporating remote sensing evapotranspiration (RS-ET) data with streamflow data for hydrological model calibration in the QTP, through a case study in the upstream region of the Ganzi Gauging station in the Yalong River Basin. Three calibration Experiments of the Soil and Water Assessment Tool model were conducted using streamflow data, RS-ET data of the Global Land Evaporation Amsterdam Model, and the combination of the both data, under the framework of the Generalized Likelihood Uncertainty Analysis (GLUE). The results show that compared with calibration using streamflow data solely, the Nash-Sutcliffe Efficiency of simulated streamflow for the calibration using both types of data increased from 0.71 to 0.81, and 0.75 to 0.84 in the calibration and validation period, respectively. Based on analysis of parameter posterior distribution, it is shown that using both types of observations could reduce the degree of equifinality. According to the Sobol' sensitivity analysis, besides the parameter related to runoff generation, some parameter regarding soil water movement, evapotranspiration and snow melting processes are also found to be sensitive in Experiment III, indicating the connection between snow melting and evapotranspiration in this high-altitude region could be captured by the model calibrated on both types of data, which explains the improvement in model performance. The findings from this study indicates that, in the QTP, integrating RE-ET data with streamflow data for calibration could improve performance of model output and more importantly, make model behavior closer to the real hydrological cycle characteristics than traditional calibration relies on streamflow data at basin outlet.

**2. Line 35, the improvement of NSE (e.g., from 0.71 to 0.81) is not significant compared with the previous studies (e.g., from 0.41 to 0.81 in Immerzeel and Droogers (2008))**

**Responses:**
For the Immerzeel and Droogers (2008), their improvement from 0.41 to 0.81 refers to the value of coefficient of determination ($R^2$) for monthly sub-basin actual ET. For our study, the improvement from 0.71 to 0.81 refer to the value of the Nash-Sutcliffe Efficiency (NSE) of simulated monthly streamflow. These two pairs of values describe the accuracy of different simulated variables using different efficiency coefficients. Therefore, it is a little difficulty to judge which one is better by comparing these values in the two paper directly. In our opinion, using NSE to quantify model performance is more effective than $R^2$.

**3. Lines 56-56, any connection with this study's topic?**

**Responses:**

Thank you very much for this comment. To better describe the scientific equation we are targeting, the sentence has been revised as follows:

In the high-altitude and data-sparse Qinghai-Tibet Plateau, streamflow data at basin outlet may be available for distributed hydrological model calibration. However, interactions among multiple internal processes of water cycle in this unique region cannot be captured by streamflow data alone, which will bring uncertainty to model simulation.

**4. Lines 276-277, why?**

**Response:**

In Experiment II, the objective of automatic calibration is to minimize the difference between simulated ET and RS-ET only. While in Experiment III, both of ET and streamflow observations are used for calibration. Therefore, there is a compensation of model performance between ET and streamflow, which may lead to the slight decrease of model performance of ET, compared with Experiment II. Similar phenomenon also was observed in the study of Herman *et al.* (2018)

**5. Figure 4b, what's the reason for the underestimation of Q between 2004 and 2005**

**Response:**

For this period, simulations of streamflow for Experiment I (calibration using streamflow solely) and Experiment III (calibration using streamflow and RS-ET) all underestimated observed values. In our opinion, this phenomenon may be caused by satellite precipitation dataset, which is used as the input of hydrological model.

**6. In Figure 8, the authors can try only to provide the key parameters in the figure to reduce reader interference.**

**Response:**

We only keep the parameters that posterior distribution is significant different from the prior distribution in either of the three Experiments. By doing this, the number of parameters being shown in the figure reduce from 28 to 10. The posterior distributions of key parameters are shown in Figure R2-R4.

[Figure]

**Figure R2.** Posterior parameter distributions of Experiment I.

[Figure]

**Figure R3.** Posterior parameter distributions of Experiment II

[Figure]

**Figure R4.** Posterior parameter distributions of Experiment III

**7. Line 346, "when combing streamflow and RS-ET data for model calibration, the accuracy of simulated streamflow and ET are all higher." I remember that the accuracy of evaporation in experiment two is higher than in experiment three (Lines 276-277).**

**Response:**

We apology for this typo. The sentence has been corrected as follow:

when combing streamflow and RS-ET data for model calibration, the accuracy of simulated streamflow is higher.

**References:**

Dembélé, M., Hrachowitz, M., Savenije, H.H.G., Mariéthoz, G., Schaefli, B.: Improving the Predictive Skill of a Distributed Hydrological Model by Calibration on Spatial Patterns With Multiple Satellite Data Sets, Water Resources Research, 56(1), doi:10.1029/2019WR026085, 2020b.

Guo, D., Yang, M., Wang H.: Characteristics of land surface heat and water exchange under different soil freeze/thaw conditions over the central Tibetan Plateau, Hydrological Processes, 25(16):1-11, doi:10.1002/hyp.8025, 2011.

Herman, M.R., Nejadhashemi, A.P., Abouali, M., Hernandez-Suarez, J.S., Daneshvar, F., Zhang, Z., Anderson, M.C., Sadeghi, A.M., Hain, C.R., Sharifi, A.: Evaluating the role of evapotranspiration remote sensing data in improving hydrological modeling predictability, Journal of Hydrology, 556, 39-49, doi:10.1016/j.jhydrol.2017.11.009, 2018.

Huang, Q., Qin, G., Zhang, Y., Tang, Q., Liu, C., Xia, J., Chiew, F.H.S., Post, D.: Using Remote Sensing Data-Based Hydrological Model Calibrations for Predicting Runoff in Ungauged or Poorly Gauged Catchments, Water Resources Research, 56(8), doi:10.1029/2020wr028205, 2020.

Immerzeel, W.W., Droogers, P.: Calibration of a distributed hydrological model based on satellite evapotranspiration, Journal of Hydrology, 349(3-4), 411-424, doi:10.1016/j.jhydrol.2007.11.017, 2008.

Kang, E.S., Cheng, G.D., Lan, Y.C., Chen, X.Z.: Alpine Runoff Simulation of the Yalong River for the South-North Water Diversion. Journal of Glaciolgy & Geocryology, 23(2), 139-148, doi:10.3969/j.issn.1000-0240.2001.02.006, 2001.

López López, P., Sutanudjaja, E.H., Schellekens, J., Sterk, G., Bierkens, M.F.P.: Calibration of a large-scale hydrological model using satellite-based soil moisture and evapotranspiration products, Hydrology and Earth System Sciences, 21(6), 3125-3144, doi:10.5194/hess-21-3125-2017, 2017.

Moazenzadeh, R., Izady, A.: A hybrid calibration method for improving hydrological systems using ground-based and remotely-sensed observations, Journal of Hydrology, 615: 128688, doi:10.1016/j.jhydrol.2022.128688, 2022.

Pianosi, F., Beven, K., Freer, J., Hall, J. W., Rougier, J., Stephenson, D. B., Wagener T.: Sensitivity analysis of environmental models: A systematic review, Environmental Modelling & Software, 79, 214-232, doi: 10.1016/j.envsoft.2016.02.008, 2017.

Willem Vervoort, R., Miechels, S.F., van Ogtrop, F.F., Guillaume, J.H.A.: Remotely sensed evapotranspiration to calibrate a lumped conceptual model: Pitfalls and opportunities. Journal of Hydrology, 519, 3223-3236, doi:10.1016/j.jhydrol.2014.10.034, 2014.

Yang, Y., Chen, R., Liu, G., Liu, Z., Wang, X.: Trends and variability in snowmelt in China under climate change, Hydrology and Earth System Sciences, 26: 305-329. doi: 10.5194/hess-26-305-2022, 2022.